Advances in solubilization and stabilization techniques for structural and functional studies of membrane proteins

Zhou Zhuanghan 1
Chen Zheng 1
Li Yiran 1
Mao Xingyue 1
Chen Junjie 1
Zhou Xuan 1
Zhang Bo boz@wku.edu.cn 1 2
1 College of Science, Mathematics and Technology, Wenzhou-Kean University , Wenzhou , Zhejiang Province , China
2 Dorothy and George Hennings College of Science, Mathematics and Technology, Kean University of New Jersey , Union , NJ , United States of America
Upadhyay Rohit
Electronic publication date: 2025 Apr 4
Publication date: 2025
Volume: 13
Electronic Location ID: e19211
Received 2024 Dec 10; Accepted 2025 Mar 5
Copyright: ©2025 Zhou et al.
Copyright year: 2025
Copyright holder: Zhou et al.
License: This is an open access article distributed under the terms of the Creative Commons Attribution License, which permits unrestricted use, distribution, reproduction and adaptation in any medium and for any purpose provided that it is properly attributed. For attribution, the original author(s), title, publication source (PeerJ) and either DOI or URL of the article must be cited.
License URL: https://creativecommons.org/licenses/by/4.0/

Keywords: Membrane protein, Membrane solubilization, Detergent-free extraction, SMA, DIBMA

Funding: Wenzhou-Kean University Research Program IRSPK202101 WKUSPF202431 This work was supported by the Wenzhou-Kean University Research Program IRSPK202101 and WKUSPF202431. The funders had no role in study design, data collection and analysis, decision to publish, or preparation of the manuscript.

==============================
Membrane proteins (MPs) are indispensable in various biological processes, including material transport, signal transduction, immune response, and cell recognition. Unraveling the intricate interplay between MP structure and function is pivotal for advancing fundamental biology and pharmaceutical research. However, the inherent hydrophobicity and complex lipid interactions of MPs pose significant challenges in determining their three-dimensional configurations. In recent years, cryo-electron microscopy (cryo-EM) has emerged as a powerful alternative for structural elucidation, overcoming the challenges faced by traditional techniques such as X-ray crystallography and nuclear magnetic resonance (NMR). This review centers on advanced solubilization and stabilization techniques for MPs, as well as MP functions and expression systems, highlighting the strengths and limitations of conventional detergents, liposomes, bicelles, and nanodiscs, alongside emerging alternatives like styrene-maleic acid (SMA) and diisobutylene-maleic acid (DIBMA). Notably, SMA and its derivatives provide promising detergent-free alternatives that preserve protein stability and native conformation, which is particularly valuable for accurate cryo-EM characterization of complex MPs. This work is designed to serve as both an updated resource for researchers already immersed in the field and an accessible entry point for those new to MP research. By consolidating recent advancements and highlighting critical gaps, this review aims to inspire future investigations that push the boundaries of MP structural and functional studies, ultimately driving innovations in drug discovery and therapeutic development.

Survey Methodology

A comprehensive and systematic search was conducted to identify relevant publications on advancements in membrane protein solubilization and stabilization techniques. Scientific databases, including PubMed, Scopus, Web of Science, and ScienceDirect, were queried to ensure broad and inclusive coverage of peer-reviewed literature. Search terms included but were not limited to “membrane protein,” “extraction,” “stabilization,” “detergents,” “nanodiscs,” “SMA,” “DIBMA,” and “detergent-free techniques.” Criteria for inclusion prioritized original research and review articles from reputable journals, emphasizing studies that addressed traditional detergents, emerging alternatives like amphipols, nanodiscs, and styrene-maleic acid (SMA)/diisobutylene-maleic acid (DIBMA) technologies, and their applications in achieving significant insights into membrane protein structure and function. Foundational works and recent advancements demonstrating practical applications or significant breakthroughs in the field were emphasized to ensure a balanced and thorough analysis. To minimize selection bias, references from key studies were manually screened to identify additional relevant publications. This methodology ensures the inclusion of high-quality, impactful studies, providing a robust basis for the synthesis presented in this review.

Introduction

Cells are the fundamental units of life, with membranes functioning as essential interfaces in both prokaryotic and eukaryotic organisms. Composed of a phospholipid bilayer featuring hydrophobic interiors and hydrophilic exteriors, these membranes act as selective barriers, regulating processes like material transport, signal transduction, immune responses, and cell recognition (Bretscher, 1985; Hegde & Keenan, 2021; Tosaka & Kamiya, 2023). Membrane proteins (MPs) are central mediators in these processes and are crucial for understanding the roles of MPs in biological and pharmaceutical research (Santos et al., 2016; Li et al., 2021).

Understanding the intricate relationship between the structure and function of MPs necessitates a thorough comprehension of three-dimensional configurations (Li et al., 2021; Wagner et al., 2024). Traditional techniques in protein structure elucidation, such as X-ray crystallography and nuclear magnetic resonance (NMR), have been instrumental in advancing our understanding of protein architectures. However, these methods often encounter limitations when applied to MPs due to their inherent hydrophobicity and complex interactions with lipid environments. Furthermore, the high molecular weight and intricate nature of MPs pose additional challenges, hindering effective structural determination. In particular, high-molecular-weight MPs and large MP complexes pose significant challenges due to their size, complexity, and dynamic nature, which further complicate structural determination (Guo et al., 2024; Wagner et al., 2024).

In recent years, cryo-electron microscopy (cryo-EM) has emerged as the leading technique for resolving MP structures (Lee et al., 2022). By enabling the visualization of MPs in a near-native state, cryo-EM has substantially enhanced the efficiency and accuracy of structural analysis. For a comprehensive overview of structure elucidation techniques, including NMR, X-ray crystallography, and cryo-EM, refer to the review articles (Piper et al., 2022; Hu et al., 2021; Kermani & Kermani, 2021; Günsel & Hagn, 2022; Boulos et al., 2023; Zadorozhnyi, Gronenborn & Polenova, 2024). Nevertheless, the successful interpretation of MP structures heavily relies on the expression, purification, and selection of appropriate solubilizing agents. Hence, this review emphasizes the diverse reagents employed for MP solubilization, encompassing conventional detergents, proteoliposomes, nanodiscs, and styrene-maleic acid lipid particles (SMALP) technology. Additionally, an overview of the classifications and roles of MPs, along with a comparative analysis of various expression systems, is provided to highlight their significance and relevance.

Classification of MPs

Based on interactions with the phospholipid bilayer and topological configurations, MPs can be categorized into three primary types: integral membrane proteins (IMPs), peripheral membrane proteins (PMPs), and lipid-anchored membrane proteins (LMPs) (Chou & Elrod, 1999; Boes, Godoy-Hernandez & McMillan, 2021). Notably, around 20–30% of genes in most genomes encode MPs (Krogh et al., 2001), and approximately 30% of naturally occurring proteins are embedded within biological membranes (Tan, Hwee & Chung, 2008).

IMPs, also known as intrinsic MPs, encompass both transmembrane and partially embedded proteins (Hegde & Keenan, 2021). The transmembrane regions of IMPs are predominantly α-helical, with their biogenesis occurring at the eukaryotic endoplasmic reticulum or through analogous processes in the prokaryotic plasma membrane. In contrast, β-barrel transmembrane domains are characteristic of the outer membranes of Gram-negative bacteria, as well as mitochondria and chloroplasts (Walther, Rapaport & Tommassen, 2009). Well-studied IMPs include G protein-coupled receptors (GPCRs), which are critical in signal transduction pathways; ion channels and solute carriers (SLCs), which regulate ion and metabolite transport; and transmembrane sensors involved in two-component systems, which play essential roles in bacterial responses to environmental changes.

In contrast, PMPs interact with the membrane through non-covalent bonds, forming transient complexes with either MPs or lipids. This dynamic group of proteins is fundamental to several cellular processes, particularly in mediating cell signaling pathways and facilitating enzymatic activities (Nastou, Tsaousis & Iconomidou, 2020; Kwan, Axford & Moraes, 2020). LMPs, meanwhile, are anchored to lipid molecules within the membrane via covalent bonds. Primarily situated on the membrane surface, these proteins are pivotal in diverse biological processes, including signal transduction and regulating material transport. The lipid anchors stabilize LMPs’ association with the membrane and facilitate interactions with other cellular components, thereby enhancing functional versatility (Ray, Jatana & Thukral, 2017).

Membrane Protein Expression Systems

A major challenge in studying MPs is obtaining sufficient quantities of the targeted proteins. Embedded in the phospholipid bilayer, MPs feature hydrophobic transmembrane regions crucial for function, thereby limiting their natural expression levels (He, Wang & Yan, 2014). To address this issue, researchers often employ heterologous overexpression strategies to enhance the yield of target MPs.

Selecting an appropriate host system and optimizing conditions are crucial. Table 1 delineates the spectrum of common expression systems, including bacteria, yeast, insect cells, plant cells, and mammalian cells, each characterized by its distinct advantages and disadvantages. For example, bacterial systems like Escherichia coli offer rapid growth and easy manipulation, but often lack the post-translational modifications needed for functional MPs, in contrast, eukaryotic systems such as yeast and mammalian cells provide environments for proper folding and modifications, albeit with higher complexity and cost (Stevens, 2000).

Table 1 Comparison of expression systems for membrane protein.

	Advantages	Disadvantages	Commonly used systems	
Bacteria	Short production cycle, easy to operate, low cost, suitable for simple proteins	Proteins may misfold, lack proper post-translational modifications (e.g., glycosylation), potential toxicity to the host	Escherichia coli, Bacillus subtilis, Lactococcus lactis, Pseudomonas putida	
Yeast	Eukaryotic system, easy to manipulate, capable of some post-translational modifications	Limited and often non-mammalian glycosylation, potential for abnormal protein structures, may not efficiently express complex proteins	Pichia pastoris, Saccharomyces cerevisiae, Schizosaccharomyces pombe, Hansenula polymorpha, Kluyveromyces lactis	
Insect cells	Suitable for large-scale production, capable of complex post-translational modifications, less prone to contamination	Requires specialized systems for post-translational modifications, different glycosylation pattern from mammals, higher cost	Drosophila melanogaster, Spodoptera frugiperda (Sf9, Sf21), Trichoplusia ni (HighFive), Bombyx mori	
Plants (whole cells or tissues)	Stable, capable of complex post-translational modifications, increasingly used in molecular farming	Long production cycle, low yield, post-translational modifications may differ from mammals, risk of immunogenicity	Arabidopsis thaliana, Oryza sativa, Nicotiana tabacum, Nicotiana benthamiana, Medicago truncatula	
Mammalian cells	Ideal for membrane proteins and complex therapeutic proteins, folding similar to mammalian systems, best for accurate post-translational modifications	High cost, long development time, scalability challenges despite mammalian-like protein folding	COS-1, HEK293, 293T, CHO, NS0, BHK-21, HeLa, Jurkat, Vero, 3T3 cells.	

Despite the availability of various expression systems, comprehensive studies that compare the production efficiency and functionality of different MPs across these systems remain limited. Consequently, careful experimental evaluation is essential when selecting an appropriate heterologous expression system for a specific MP.

MP Extraction and Stabilization

A typical MP extraction and purification process involves solubilizing the membrane fraction, followed by stabilizing and purifying the MP in a membrane-like environment. Detergents are commonly used to enhance MP solubility, while affinity tags aid in purification. To preserve functionality, MPs are stabilized using artificial lipids, such as those in proteoliposomes or nanodiscs, which mimic the natural membrane environment. Over the past decade, the styrene-maleic acid lipid particles (SMALP) technique, which leverages an amphiphilic copolymer to stabilize MPs with native cell membrane lipids, has gained widespread popularity and shows great promise as a mainstream approach in MP research (Knowles et al., 2009; Lee et al., 2016).

Conventional detergents

Detergents, or surfactants, are amphiphilic molecules with polar hydrophilic heads and non-polar hydrophobic tails, mirroring the architecture of biological membrane phospholipids (Fig. 1A). In aqueous solutions, detergents spontaneously form micelles once they reach the critical micelle concentration (CMC) (Ruckenstein & Nagarajan, 1981). At concentrations above the CMC, detergents disrupt membranes and solubilize MPs, forming protein-detergent complexes that stabilize the proteins. Generally, lower CMC values suggest greater micelle stability and a reduced risk of protein denaturation.

Figure 1 Techniques for membrane protein (MP) solubilization and stabilization.

(A) Protein-detergent complex: the MP is centrally located, surrounded by detergent molecules (white) that stabilize the hydrophobic regions. (B) Amphiphilic polymer encapsulation: the MP is embedded within an amphiphilic polymer matrix, with hydrophobic regions (dark green) and hydrophilic regions (orange) of the polymer aiding in stabilization. (C) Lipopeptide detergents (LPDs): the MP is positioned centrally, flanked by structured lipopeptides (orange) that provide support and mimic the membrane environment. (D) Liposomes: the MP is integrated within a spherical phospholipid bilayer (white), offering a near-native lipid environment. (E) Bicelles: the MP is stabilized within a bilayer-like structure formed by a mix of long-chain phospholipids (yellow) and short-chain detergents or lipids (gray), creating a flat, disc-shaped bilayer with detergent around the edges. (F) Nanodiscs: the MP is surrounded by a phospholipid bilayer (yellow) and stabilized by membrane scaffold proteins (MSP, green). (G) Styrene-maleic acid lipid particles (SMALPs): the MP is encapsulated within native lipid bilayers (yellow), stabilized by SMA copolymers (green). (H) Diisobutylene-maleic acid lipid particles (DIBMALPs): the MP is enclosed in a phospholipid bilayer (yellow), stabilized by DIBMA copolymers (green).

Another key factor is the hydrophile-lipophile balance (HLB), determined by the ratio of hydrophilic to hydrophobic groups and correlates with a detergent’s ability to solubilize MPs (Hirama & Torii, 2015). Additionally, the aggregation number, representing the average number of detergent monomers in a micelle, is crucial, as excessive aggregation may result in undesirable protein aggregation within micelles.

Detergents such as n-dodecyl-β-D-maltopyranoside (DDM) and n-octyl-β-D-glucopyranoside (OG) have been instrumental in the extraction, solubilization, and structural analysis of MPs, enabling high-resolution crystallization of α-helical proteins. However, these detergents exhibit limitations, including the formation of large micelles that obscure protein structures, slow micelle tumbling that degrades NMR spectroscopy quality, and the requirement for high concentrations to maintain protein stability (Stetsenko & Guskov, 2017). In Table 2, commonly employed surfactants are categorized based on their chemical composition and inherent properties.

Table 2 Types of detergents for membrane protein extraction.

Type	Chemical structure	Advantages	Disadvantages	Examples	
Ionic	Hydrophilic head carries a positive or negative charge	Strong extraction ability	High polarity may cause protein denaturation or inactivation	Sodium dodecyl sulfate (SDS), sodium deoxycholate, sodium taurocholate	
Non-Ionic	Hydrophilic head carries no charge	Good optical properties; useful for monitoring protein stability and concentration during purification	Weaker extraction ability	n-Octyl-β-D-glucopyranoside (OG), n-Decyl-β-D-maltopyranoside (DM), n-Dodecyl-β-D-maltopyranoside (DDM), polyethylene glycol derivatives (e.g., C12E8), Triton X-100, Tween 20, Brij 35	
Zwitterionic	Hydrophilic head carries both positive and negative charge	Gentle extraction with minimal protein denaturation; preserves native protein state	Sensitive to strong ionic environments, may alter protein function	CHAPS, CHAPSO, LDAO, sodium lauroyl sarcosinate, styrene-maleic acid copolymer (SMA), diisobutylene-maleic acid (DIBMA)	

Improvements in detergent technologies

Maintaining the functional integrity of MPs during extraction and purification is challenging due to their dependence on specific interactions with native lipids. While effective in solubilizing MPs, conventional detergents can disrupt these lipid-protein interactions, potentially leading to denaturation or loss of function. Beyond serving as fundamental building blocks of cellular membranes, lipids also modulate protein function and structural conformation (Saliba, Vonkova & Gavin, 2015).

To mitigate these issues, recent advancements in detergent design have led to the development of more stabilizing alternatives. Among these, lauryl maltose neopentyl glycol (LMNG) has been widely adopted due to its ability to maintain the homogeneity of MP complexes, making it particularly effective for the structural analysis of GPCRs (Breyton et al., 2019; Lee et al., 2022). Similarly, glyco-diosgenin (GDN), a steroid-based detergent featuring a diosgenin backbone, has demonstrated superior stabilization of fragile MPs compared to traditional detergents like DDM (Chae et al., 2012; Smirnova, Wu & Brzezinski, 2025). The hydrophobic steroid moieties in these detergents contribute to their ability to preserve the structural integrity of MPs that are otherwise unstable in conventional detergents (Lee et al., 2022). However, the limited synthetic accessibility of these detergents has led to the development of glyco-steroidal amphiphiles (GSAs), which incorporate cholestanol, cholesterol, and diosgenin as promising alternatives to GDN (Ehsan et al., 2022).

As detergent-based approaches continue to evolve, alternative surfactant technologies have also been developed to provide enhanced MP stabilization in detergent-free environments.

Amphipols: novel surfactants for MP solubilization

Amphipols are amphiphilic polymers designed to solubilize MPs by forming stable complexes without the use of conventional detergents (Zoonens & Popot, 2014). Their amphiphilic nature arises from a comb-like structure, with hydrophilic and hydrophobic groups distributed along a polymer backbone, mimicking the lipid environment of membranes (Fig. 1B). Unlike conventional surfactants that can surround the entire protein with micelles, amphipols coat only the hydrophobic regions of the protein, thus preserving its overall structure and function.

One of the most widely studied amphipols, A8-35, is polyacrylic acid modified with 35% hydrophobic octyl and isopropyl side chains (Le Bon et al., 2018), effectively providing a pseudo-lipid environment for MPs. This configuration stabilizes MPs by mimicking the native acyl chains of phospholipids, preventing aggregation and preserving protein function in aqueous solutions. A8-35 has been successfully applied to stabilize and solubilize a variety of MPs, such as bacteriorhodopsin and cytochrome b6f (Ratkeviciute, Cooper & Knowles, 2021). Amphipols have also been used with several GPCRs, maintaining their native conformations and ligand-binding capabilities (Jones et al., 2020).

Lipopeptide detergents (LPDs): membrane-mimicking polypeptides

Lipopeptide detergents (LPDs) represent a new class of amphiphilic molecules specifically designed to mimic the bilayer arrangement of MPs’ hydrophobic surfaces. Each LPD is composed of an α-helical peptide (Privé, 2009) or a β-strand (Gurung et al., 2024) backbone, with alkyl chains anchored to both ends of the peptide (Fig. 1C). Unlike conventional detergents, which form disordered micelles that can destabilize MPs, LPDs create more organized structures with reduced curvature when interacting with phosphatidylcholine bilayers.

Currently, LPDs remain largely experimental and have yet to see widespread adoption in structural studies. One fundamental limitation is the complexity and high cost of designing and synthesizing peptide backbones tailored to specific MP applications (Privé, 2009; Veith et al., 2017; Gurung et al., 2024). Despite these challenges, ongoing research aims to refine LPDs and extend their use in maintaining the native conformations of MPs. Additionally, novel alternatives such as beltides (Larsen et al., 2016) and peptergents (Yeh et al., 2005) have emerged, offering more flexibility and adaptability than LPDs.

Liposomes and bicelles: providing native-like lipid environments

Liposomes, spherical lipid vesicles that form spontaneously in aqueous solutions, offer a lipid environment that preserves the characteristic shape and function of MPs (Bangham, 1995). However, since MPs cannot be directly incorporated into preformed liposomes, they must first be solubilized with detergents and then reconstituted into protein-lipid complexes. Detergent removal techniques, such as biobeads or dialysis, are employed to lower the detergent concentration below the CMC, enabling the formation of functional protein-lipid assemblies (Rigaud & Lévy, 2003; Warschawski et al., 2011).

Liposomes are versatile tools for studying MPs, providing a stable, native-like lipid environment (Fig. 1D). This makes them superior to other membrane mimetics. In mass spectrometry, liposomes aid in retaining intact protein-lipid complexes, facilitating the study of protein-lipid interactions (Frick, Schwieger & Schmidt, 2021). Furthermore, they are also invaluable in cryo-EM studies, as they preserve membrane curvature and electrochemical gradients, facilitating the analysis of complex proteins like AcrB—a ∼350 kDa homo-trimer multidrug transporter from E. coli that plays a crucial role in multidrug resistance, resolved at 3.9 Å (Yao, Fan & Yan, 2020).

Bicelles, on the other hand, form small, bilayer-like structures after detergent treatment, mimicking biological membranes with greater stability (Fig. 1E) (Poulos et al., 2015). Bicelles are particularly advantageous in NMR studies due to their ability to align in magnetic fields, as demonstrated with the bacteriophage Pf1 coat protein (Dufourc, 2021). Bicelles also support solution-state NMR, preserving native protein folding and dynamics, as seen with cytochrome b 5 (important in electron transport and fatty acid metabolism) (Dürr et al., 2007; Dürr, Gildenberg & Ramamoorthy, 2012). Moreover, bicelles also facilitate the investigation of protein-lipid interactions, exemplified by the study of mastoparan X, a wasp venom peptide known for its membrane-disrupting properties (Whiles et al., 2001).

Nanodiscs: addressing limitations of liposome and peptide-based detergents

Nanodiscs have emerged as a viable solution to the complexities associated with preparing liposomes and peptide-based detergents, which can often be unstable and lack reproducibility. Nanodiscs consist of a circular lipid bilayer stabilized by membrane scaffold proteins (MSPs), typically derived from human high-density lipoprotein particles or engineered apoA-I variants (Sligar & Denisov, 2021). When MSPs, lipids, and detergents are mixed with the target MP, they spontaneously self-assemble into disc-shaped complexes (Fig. 1F). Subsequent removal of the detergent leaves the MP stabilized within a lipid environment that closely mimics its native state (Denisov & Sligar, 2017).

Despite advantages, nanodiscs present certain limitations, including challenges in controlling disc size and the continued necessity for detergents during assembly (Sligar & Denisov, 2021; Krishnarjuna & Ramamoorthy, 2022). To address these issues, covalently circularized nanodiscs (cNDs) were developed. Unlike conventional nanodiscs, cNDs feature a covalent linkage between the N- and C-termini of MSPs, yielding a circular structure that enhances stability, size homogeneity, and proteolytic resistance (Nasr et al., 2016; Nasr & Wagner, 2018). These properties make cNDs particularly advantageous for studying large MP complexes, such as mammalian respiratory complex I and pore-forming toxins (Nasr et al., 2016; Nasr & Wagner, 2018). Moreover, recent advancements, including SpyCatcher-SpyTag and microbial transglutaminase (MTGase)-based strategies, have further optimized cND stability and expanded the range of achievable nanodisc sizes from ∼8.5 nm to 50 nm (Zhang et al., 2021; Dong et al., 2024).

In addition to cNDs, peptidiscs provide another detergent-free alternative for MP stabilization. Unlike nanodiscs, which require lipids, peptidiscs rely on short amphipathic bi-helical peptides that spontaneously wrap around the MP, shielding its hydrophobic surfaces without the need for additional lipids (Carlson et al., 2018). This approach is simple, cost-effective, and compatible with high-throughput purification protocols, making it attractive for structural and functional studies (Jandu et al., 2024). Peptidiscs have successfully stabilized E. coli proteins such as MsbA and MscS for cryo-EM visualization (Angiulli et al., 2020) and have also been applied to generate membrane proteome libraries from mammalian cells (Zhao et al., 2023; Antony et al., 2024).

Beyond these advancements, the saposin-lipoprotein nanoparticle (Salipro) system offers another detergent-free strategy for MP stabilization (Frauenfeld et al., 2016). This approach is particularly effective for challenging MPs, such as SLC transporters, chemokine receptor (CKR) family GPCRs, and the E. coli ABC transporter hemolysin B (HlyB), which are generally difficult to maintain in functional states with conventional methods (Kanonenberg, Smits & Schmitt, 2019; Lloris-Garcerá et al., 2020). Additionally, polymer-based nanodiscs have been developed to enable direct MP extraction from native membranes while offering improved control over disc size (Knowles et al., 2009; Brown, Trieber & Overduin, 2021).

SMALP technology: detergent-free MP stabilization

SMALP technology was developed as a detergent-free alternative to traditional nanodiscs, eliminating the need for surfactants and MSPs (Fig. 1G). SMALP uses the amphiphilic copolymer styrene-maleic acid (SMA) to directly stabilize MPs by encapsulating them with native membrane lipids, forming particles approximately 10 nm in diameter (Knowles et al., 2009).

Early applications of SMALP involved the addition of artificial lipids and MSPs, but subsequent research demonstrated that SMA could retain native membrane lipids without external additives. Although the precise self-assembly mechanism remains an area of debate, SMALP is generally regarded as a detergent-free process. Some studies, however, suggest that SMA may behave similarly to a detergent while simultaneously forming nanodisc-like structures (Xue et al., 2018; Kamilar et al., 2023). This dual behavior means the concentration and type of SMA must be carefully optimized for successful MP extraction.

SMA is synthesized through free-radical styrene and maleic anhydride polymerization, followed by alkaline hydrolysis to form the functional copolymer (Lee et al., 2016). This process is less time-consuming and more cost-effective compared to synthesizing many other membrane-stabilizing agents. The molecular weight of SMA is typically below 10 kDa, with a styrene-to-maleic acid ratio between 2:1 and 3:1 (Morrison et al., 2022). This ratio directly influences solubilization efficiency, making SMA composition optimization critical for MP stabilization.

SMALP technology has revolutionized MP research by preserving the native lipid environment during extraction, thereby enhancing the structural and functional integrity of complex MPs (Hesketh et al., 2020). For instance, SMA facilitated the extraction and 3.2-Å resolution imaging of the AcrB trimer, revealing its interactions with associated lipid bilayers around its transmembrane domains (Qiu et al., 2018). Additionally, SMA enabled the extraction of a 464 kDa ACIII–cyt aa 3 supercomplex from Flavobacterium johnsoniae, yielding high-resolution cryo-EM data that illuminated electron transport dynamics within the respiratory chain (Sun et al., 2018). SMA also proved effective for isolating low-expressing GPCRs, preserving functionality and capturing the binding of the neurotransmitter neurotensin to the D1 receptor, a crucial target in pharmaceutical research (Bada Juarez et al., 2020).

Despite its advantages, SMA is sensitive to low pH and divalent cations like calcium and magnesium, which can induce precipitation and restrict its use under certain experimental conditions (Knowles et al., 2009). Another concern is that SMA may over-stabilize MPs, potentially limiting the conformational flexibility required for proper function (Pollock et al., 2018). Moreover, SMA’s strong UV absorbance at 260 nm complicates optical spectroscopic analyses, making it difficult to accurately quantify solubilized proteins (Lee et al., 2016; Oluwole et al., 2017). Given these challenges, optimizing SMA for different MPs is often a protein-specific task, requiring extensive trial and error to establish effective extraction protocols.

Modifications to SMA: developing calcium-, magnesium-, and pH-stable alternatives

Recent innovations have led to the development of diisobutylene-maleic acid (DIBMA), a copolymer of maleic acid and diisobutylene (2,4,4-trimethylpent-1-ene), as an alternative to the styrene-based SMA (Fig. 1H) (Oluwole et al., 2017). DIBMA exhibits greater stability across a broader pH range and demonstrates superior resistance to ion-induced precipitation, particularly from divalent cations such as calcium and magnesium. It also circumvents the intense UV absorbance associated with SMA, thereby simplifying spectroscopic analyses and enhancing compatibility with various experimental conditions (Gulamhussein et al., 2020). Furthermore, DIBMA solutions are easier to prepare, requiring only basic dialysis, bypassing the labor-intensive precipitation, washing, and lyophilization steps required for SMA (Rothnie, 2016).

However, DIBMA nanodiscs tend to be less stable than SMALPs, likely due to their larger size and higher lipid content. This reduced stability can be a drawback for certain proteins, but the trade-off may also yield benefits. The more dynamic, native-like lipid environment of diisobutylene-maleic acid lipid particles (DIBMALPs) could better accommodate protein conformational changes, as well as thermodynamic and kinetic studies, compared to the more rigid structure provided by SMA (Gulamhussein et al., 2020). Optimizing the concentration and composition of DIBMA remains crucial, as improper ratios can lead to inefficient solubilization or destabilization of MPs, echoing the trial-and-error processes often encountered with SMA-based techniques (Dimitrova et al., 2022).

To further address the limitations of traditional SMA-based nanodiscs, styrene maleic acid-quaternary ammonium (SMA-QA) has been developed. SMA-QA is synthesized by modifying SMA with aminoethyltrimethylammonium chloride hydrochloride, replacing pH-sensitive carboxylate groups with permanently charged quaternary ammonium groups (Ravula et al., 2018). This modification enhances stability across a broad pH range (2.5–10) and high divalent metal ion concentrations. Additionally, precise size control (10–30 nm) is achieved, producing monodisperse nanodiscs suitable for NMR studies (Ravula et al., 2018; Ravula, Hardin & Ramamoorthy, 2019). Compared to standard SMA (2:1), improved extraction efficiency, functional stability, and preservation of a more native-like lipid environment have been demonstrated (Zhang et al., 2023). Other SMA derivatives, including SMA with solvent-exposed sulfhydryls (SMA-SH), styrene maleic anhydride–ethanolamine (SMA-EA), styrene maleic acid–ethylenediamine (SMA-ED), styrene maleimide–amine (SMAd-A), and zwitterionic SMA, have also shown promising potential for MP stabilization and structural studies (Lindhoud et al., 2016; Fiori et al., 2017; Ravula et al., 2018).

In addition to SMA modifications, novel polymer-based nanodiscs have been explored, with inulin-based nanodiscs emerging as a promising non-ionic alternative (Ravula & Ramamoorthy, 2021; Krishnarjuna et al., 2022). These nanodiscs are constructed using amphiphilic polymers derived from hydrophobically functionalized inulin, a naturally occurring fructo-oligosaccharide. Unlike SMA-derived systems, charge neutrality is maintained, preventing undesirable electrostatic interactions and improving compatibility with ion-exchange chromatography. High stability has been observed across a pH range of 2.5–8.5, and solubility is retained in up to 100 mM Ca2+ or Mg2+, effectively overcoming the precipitation issues commonly associated with SMA-based nanodiscs (Ravula & Ramamoorthy, 2021; Krishnarjuna et al., 2022).

Conclusions

MPs are central to a wide array of biological processes, making them a focal point in research. Advances in cryo-EM have significantly enhanced the study of MP structures, yet the extraction and stabilization of these proteins continue to present major challenges. Traditional methods often fail to preserve the structural and functional integrity of MPs. The advent of nanodiscs and SMALPs, which are detergent-free alternatives that maintain native lipid environments, has revolutionized MP extraction.

Despite these advancements, several limitations persist. Future endeavors to address these challenges—such as enhancing the stability and versatility of SMALP systems or refining SMA-derived methodologies—will be essential for broadening the applications of these technologies. Such improvements could facilitate the investigation of more complex MPs, especially those with larger sizes or intricate functional mechanisms, thereby opening new avenues for understanding their roles in biological systems.

Additional Information and Declarations

Competing Interests

Author Contributions

Data Availability

The authors declare there are no competing interests.

Zhuanghan Zhou conceived and designed the experiments, performed the experiments, analyzed the data, prepared figures and/or tables, authored or reviewed drafts of the article, and approved the final draft.

Zheng Chen conceived and designed the experiments, performed the experiments, authored or reviewed drafts of the article, and approved the final draft.

Yiran Li conceived and designed the experiments, performed the experiments, analyzed the data, prepared figures and/or tables, authored or reviewed drafts of the article, and approved the final draft.

Xingyue Mao conceived and designed the experiments, performed the experiments, analyzed the data, authored or reviewed drafts of the article, and approved the final draft.

Junjie Chen conceived and designed the experiments, authored or reviewed drafts of the article, and approved the final draft.

Xuan Zhou conceived and designed the experiments, performed the experiments, analyzed the data, prepared figures and/or tables, authored or reviewed drafts of the article, and approved the final draft.

Bo Zhang conceived and designed the experiments, performed the experiments, analyzed the data, prepared figures and/or tables, authored or reviewed drafts of the article, and approved the final draft.

The following information was supplied regarding data availability:

This is a literature review.

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
