# Peer review of "Advances in solubilization and stabilization techniques for structural and functional studies of membrane proteins"

_PeerJ, doi:10.7717/peerj.19211_

## Round 0.1 · original submission · Major Revisions

Please address all reviewers' comments in the revised manuscript and include a detailed, point-by-point response.

Reviewer 1 ·

Basic reporting

The manuscript is very well written, providing an easy to understand introduction to the techniques used for the stabilisation of membrane-proteins. No issues were noted with the English used which h is appropriate to the field and to a general reader.

As a review article it presents appropriate literature predominantly from the last 15 years. While focussing primarily on the techniques used to solubilise and stabilise membrane proteins the authors do a good job of linking each to the various protein structure elucidation techniques (cryoEM, X-ray crystallisation & NMR). The review presents a broad discussion of the techniques available and their development over time however, with limited technical detail. This provides an excellent entry point for the reader to further investigate techniques that may be of interest and hence provides a good context within the field.

The introduction is clear, with a well-developed motivation and target audience. Although it is indicated that the review provides an updated, resources for those in the field, it will be of particular benefit to those that are new to membrane protein.

Experimental design

The study design appears to be robust in locating research of interest. It presents a broad range of outputs from different groups over an extended period of time. The sources presented are appropriately cited. The review is well structured, providing an initial overview before discussing individual techniques in a chronological order.

Validity of the findings

While presenting the different techniques for membrane protein solubilisation, the authors have carefully presented both the pros but also the cons of each technique. This will be particularly useful foe a person new to the field. The manuscript could have been strengthened by providing some more depth on which technique is suitable for each protein structure elucidation techniques (cryoEM, X-ray crystallisation & NMR). Or perhaps by providing some examples of proteins that were successfully studied following solubilisation by a particular technique perhaps in an additional table.

The figure presented shows the models of membrane proteins and their solubilisation/stabilisation in a simplistic but easy to follow fashion. Particular attention to panel B is required as there is poor contrast between the orange and grey, representing the hydrophilic and hydrophobic regions, when the figure is printed. It would be useful to review the other panels, as well, as the gray is often difficult to discern.

Reviewer 2 ·

Basic reporting

The manuscript by Zhou et. al. is intended to provide a broad overview of methods used to stabilize membrane proteins for structural and biochemical analysis. This topic is of broad interest, due to the widely documented challenges associated with studying membrane proteins.

However, there are numerous issues with the current version which must be addressed prior to publication. I elaborate on these points below (see details in the "additional comments" blox)

Experimental design

Many pertinent areas in the field are not adequately covered in this review or else are missing entirely (details in the "additional comments" box). Numerous statements are not adequately cited. I am surprised to see only ~55 references cited; for a review article, I would expect rather more references - the majority of which should be within the last 5 years.

Validity of the findings

No novel findings reported, this is a review article.

Additional comments

I have a number of concerns the authors must address prior to publication of this work:

1. Intro, lines 55-68: none of these statements are supported by citations. Please add appropriate references.

2. Lines 66-68: false statement. What do the authors mean by "high molecular weight and intricate nature of membrane proteins"? Many small membrane proteins (~25 kDa) have been crystallized and reported in the PDB. Please explain better.

3. Line 70-71: "By preserving the native conformation of MPs, cryo-EM has substantially enhanced the efficiency and accuracy of structural analysis". This statement is incorrect. CryoEM alone does NOT stabilize membrane protein conformations. The stabilization would have occurred during the sample preparation step, prior to EM analysis. Please correct this.

4. "Classification of MPs" line ~80: The authors appear to focus on alpha-helical membrane proteins, while completely ignoring beta-barrel proteins. Beta-barrel proteins exist in the outer membrane of Gram negative bacteria as well as in chloroplasts and mitochondria. They are essential for numerous physiological processes and must be described here in more detail.

5. Lines 86-89: "Well-studied IMPs include G protein-coupled receptors (GPCRs), critical in signal transduction pathways, and transmembrane sensors involved in two-component systems, which play essential roles in bacterial responses to environmental changes". While correct, this statement is incomplete. What about ion channels? Solute carriers (SLCs)? Transporters and channels? All of these classes of membrane proteins are extensively studied, with many PDB structures and published papers. The authors should discuss these here.

6. "MP extraction and Stabilization" (line 116): Why not present a schematic showing a typical membrane protein expression/purification workflow? Having a graphic here would capture the readers' attention and enhance the readability of the manuscript.

7. Line 138 (and Table 2): the authors have ignored many recent developments in the detergent field. I highlight in particular the detergents LMNG and GDN which have been used many times in recent years to extract fragile (often eukaryotic) membrane proteins from biological membranes for structural analysis. The authors should mention these new developments in the section beginning on Line 146.

8. Line 170: It's unclear why so much text is devoted to "lipopeptide detergents". By the author's own admission, this area is rather niche and these methods have not been widely adopted by the field.

9. Line 182: peptidiscs are NOT comparable to a lipopeptide detergent; they are more analogous to a nanodisc. Please correct this.

10. Line 209, nanodiscs: the authors discuss MSP nanodiscs and the Salipro technology in this section. They should also include a description of peptidiscs - an emerging nanodisc alternative (first described in 2018) that is gaining traction in the membrane protein structural biology field.

11. Line 219, controlling nanodisc size: there have been several papers by different groups in recent years describing covalently circularized nanodiscs of well-defined size. The authors should mention these latest developments here.

12. Figures: I am surprised a review on such a broad topic contains only a single figure. Figure 1 appears very simplistic and does not convey much information. Please redesign. Numerous reviews have been published on this topic over the years, please see some of those for "inspiration" on figure design.

Reviewer 3 ·

Basic reporting

The current review article summarizes the literature on membrane mimetics used to study membrane proteins, with a primary focus on the solubilization and stabilization techniques available in the current body of work. Overall, the review is well-written and could be published after minor revisions incorporating the following points.
The review lacks key developments in polymers designed to address specific limitations. For example, in line 267, the section discussing how polymer nanodisc technology overcomes issues like metal ion sensitivity and pH instability does not mention significant advancements, such as the SMA-QA polymer developed in Prof. Ramamoorthy's lab.
Recent advancements in non-ionic polymer nanodiscs, which address the high charge density associated with charged polymers, are not discussed. Examples include inulin-derivatives, z-SMA, and similar innovations.
Figure 1: Add the specific names of the nanodiscs in Figure 1 to enhance clarity and make it easier for readers to understand.

Experimental design

N/A

Validity of the findings

N/A

Additional comments

N/A

---

## Round 0.2 · accepted · Accept

Authors have addressed all of the reviewers' comments and manuscript is ready for publication.

Reviewer 2 ·

Basic reporting

No comment.

Experimental design

No comment

Validity of the findings

No comment

Additional comments

I thank the authors for their hard work, they have responded well to my main concerns. The current version of the manuscript is very much improved! I am happy to recommend publication "as is".

Reviewer 3 ·

Basic reporting

All of my queries have been fully addressed by the authors. I recommend this manuscript for publication in its current form.

Experimental design

NA

Validity of the findings

NA

Additional comments

NA